# Characteristics, Treatment Strategies and Outcome in Cardiogenic Shock Complicating Acute Myocardial Infarction: A Contemporary Dutch Cohort

**DOI:** 10.3390/jcm12165221

**Published:** 2023-08-10

**Authors:** Elma J. Peters, Sanne ten Berg, Margriet Bogerd, Marijke J. C. Timmermans, Adriaan O. Kraaijeveld, Jeroen J. H. Bunge, Koen Teeuwen, Erik Lipsic, Krischan D. Sjauw, Robert-Jan M. van Geuns, Admir Dedic, Eric A. Dubois, Martijn Meuwissen, Peter Danse, Niels J. W. Verouden, Gabe Bleeker, José M. Montero Cabezas, Irlando A. Ferreira, Annemarie E. Engström, Wim K. Lagrand, Luuk C. Otterspoor, Alexander P. J. Vlaar, José P. S. Henriques

**Affiliations:** 1Heart Center, Department of Cardiology, Amsterdam University Medical Centers, 1105 AZ Amsterdam, The Netherlands; e.j.peters@amsterdamumc.nl (E.J.P.);; 2Netherlands Heart Registration, 3511 EP Utrecht, The Netherlands; marijke.timmermans@nhr.nl; 3Department of Cardiology, Utrecht University Medical Center, 3584 CX Utrecht, The Netherlands; a.o.kraaijeveld-3@umcutrecht.nl; 4Department of Cardiology, Thoraxcenter, Erasmus Medical Center, 3015 GD Rotterdam, The Netherlands; j.bunge@erasmusmc.nl (J.J.H.B.);; 5Department of Intensive Care Adults, Erasmus Medical Center, 3015 GD Rotterdam, The Netherlands; luuk.otterspoor@catharinaziekenhuis.nl; 6Heart Center, Department of Interventional Cardiology, Catharina Hospital Eindhoven, 5623 EJ Eindhoven, The Netherlands; koen.teeuwen@catharinaziekenhuis.nl; 7Department of Cardiology, University Medical Center Groningen, 9713 GZ Groningen, The Netherlands; 8Heart Center, Medical Center Leeuwarden, 8934 AD Leeuwarden, The Netherlands; 9Department of Cardiology, Radboud University Medical Center, 6525 GA Nijmegen, The Netherlands; robertjan.vangeuns@radboudumc.nl; 10Department of Cardiology, Noordwest Clinics, 1815 JD Alkmaar, The Netherlands; 11Department of Cardiology, Amphia Hospital, 4818 CK Breda, The Netherlands; 12Department of Cardiology, Rijnstate Hospital, 6815 AD Arnhem, The Netherlands; 13Department of Cardiology, Haga Hospital, 2545 AA The Hague, The Netherlands; 14Department of Cardiology, Leiden University Medical Center, 2333 ZA Leiden, The Netherlands; 15Department of Cardiology, Isala Hospital, 8025 AB Zwolle, The Netherlands; 16Department of Intensive Care, Amsterdam University Medical Centers, 1105 AZ Amsterdam, The Netherlands; a.e.engstrom@amsterdamumc.nl (A.E.E.);; 17Department of Intensive Care, Catharina Hospital, 5623 EJ Eindhoven, The Netherlands

**Keywords:** cardiogenic shock, acute myocardial infarction, percutaneous coronary intervention, mortality, evidence-based therapy

## Abstract

Cardiogenic shock (CS) complicating acute myocardial infarction (AMI) is associated with high morbidity and mortality. Our study aimed to gain insights into patient characteristics, outcomes and treatment strategies in CS patients. Patients with CS who underwent percutaneous coronary intervention (PCI) between 2017 and 2021 were identified in a nationwide registry. Data on medical history, laboratory values, angiographic features and outcomes were retrospectively assessed. A total of 2328 patients with a mean age of 66 years and of whom 73% were male, were included. Mortality at 30 days was 39% for the entire cohort. Non-survivors presented with a lower mean blood pressure and increased heart rate, blood lactate and blood glucose levels (*p*-value for all <0.001). Also, an increased prevalence of diabetes, multivessel coronary artery disease and a prior coronary event were found. Of all patients, 24% received mechanical circulatory support, of which the majority was via intra-aortic balloon pumps (IABPs). Furthermore, 79% of patients were treated with at least one vasoactive agent, and multivessel PCI was performed in 28%. In conclusion, a large set of hemodynamic, biochemical and patient-related characteristics was identified to be associated with mortality. Interestingly, multivessel PCI and IABPs were frequently applied despite a lack of evidence.

## 1. Introduction

Cardiogenic shock (CS) is a clinical syndrome characterized by hypotension and end-organ hypoperfusion. Even though CS complicates only 3–13% of acute myocardial infarctions (AMI), it is the leading cause of death for patients with an acute coronary syndrome [1,2,3]. While overall 30-day mortality in AMI is around 6% in EU countries, mortality rates for AMI complicated by CS (AMICS) are as high as 40–50% [4,5,6,7].

In order to improve outcomes for AMICS patients, it is important to gain accurate insight into in-depth patient characteristics, current clinical management strategies and outcomes in this specific population. Data regarding these features are limited and often based on clinical trial data or diagnosis codes. In addition, only a few databases have been designed for CS to capture more in-depth variables. 

The primary aim of this study was to gain insights into contemporary trends in patient characteristics, current treatment strategies and outcome for AMICS patients undergoing PCI in the Netherlands. Additional aims were to investigate differences in outcome in predefined subgroups and to explore whether the current clinical practice is consistent with available treatment guidelines.

## 2. Materials and Methods

### 2.1. Patient Selection

Baseline, procedural and outcome data from all patients undergoing PCI in the Netherlands are prospectively registered with the Netherlands Heart Registration (NHR; www.nhr.nl) [8]. Relevant variables and their definitions as collected in the NHR are shown in Appendix A, Table A1. All patients with CS undergoing PCI for AMI between January 2017 and September 2021 were subsequently identified in the NHR database. Cardiogenic shock was defined as the presence of hypotension (systolic blood pressure ≤ 90 mmHg for at least 30 min or the need for supportive measures to maintain systolic blood pressure ≥ 90 mmHg) with signs of hypoperfusion of end-organs (cold extremities and/or oliguria < 30 mL/h and/or heart rate ≥ 60 beats per minute). An additional set of variables was established to be collected in patients with CS. This additional data collection was executed in 14 of 30 PCI centers in the Netherlands. See Appendix B, Table A2 for the participating hospitals.

### 2.2. Variable Selection

A draft version of the set of additional variables to be collected in patients with CS was established in consultation with interventional cardiologists and intensivists from participating hospitals. After pilot testing of this draft version, the updated version was discussed in a multidisciplinary team. A few adjustments were made prior to finalizing the selection and its corresponding data dictionary. More details of the process and the final set of variables can be found in Appendix C, Figure A1.

### 2.3. Data Collection

Clinical data for all patients were retrieved from the electronic health records. Survival status was retrieved from the governmental Personal Records Database (in Dutch: *Basisregistratie Personen*) in all hospitals with a follow-up period of at least one year. Data collection was performed by trained data managers and medical doctors with supervision by an interventional cardiologist or a cardiac intensivist. To ensure quality, several automated quality controls were carried out after data submission according to the quality control system of the NHR as described elsewhere [9]. The data were pseudonymized and locked after preliminary findings were submitted to the respective hospital with the opportunity for reviewing and complementing.

### 2.4. Statistical Analysis

Statistical analysis was performed using IBM SPSS 28.0 (IBM, SPSS, Inc., Chicago, IL, USA). Normally distributed data were displayed as mean ± standard deviation (SD) and compared in survivors and non-survivors using the unpaired *t*-test. Non-normally distributed data were described as median with interquartile range (IQR) and compared with the Mann–Whitney U test. Categorical data were displayed as frequencies and percentages and compared using the chi-square test. Temporal trends were analyzed using the Mann–Kendall test. Survival curves were constructed using the Kaplan–Meier method, and comparisons between subgroups were made with the log-rank statistic. Subgroup analyses were performed for sex (male/female), out-of-hospital cardiac arrest (OHCA) (yes/no), indication of PCI (ST-elevation myocardial infarction [STEMI]/non-ST-elevation myocardial infarction [NSTEMI]) and multivessel PCI within multivessel disease (yes/no). A *p*-value < 0.05 was considered statistically significant for all analyses. Missing data were not imputed for the current analyses. Denominators were notated for categorical variables with missing data.

## 3. Results

### 3.1. Patient Characteristics

From January 2017 to September 2021, a total of 2328 patients with AMI complicated by CS and treated with PCI were identified. This was 2.4% of the total PCI population in the selected hospitals (*n* = 98.721). The mean age was 66.4 (±12.3) years, and 72.9% of patients (*n* = 1685) were male. In this cohort, the prevalence of diabetes was 20.8% (*n* = 459), mostly treated with medication only. A total of 631 patients (29.3%) experienced a prior coronary event, most commonly a prior myocardial infarction (*n* = 482, 21.4%). Patients with CS more often presented with STEMI than with NSTEMI (86.1% vs. 13.9%, *p* < 0.001), and for most patients (*n* = 1166, 58.6%), the onset of symptoms was less than 3 hours before presentation. Of all patients, 934 (40.3%) presented after an OHCA. Details on patient characteristics are displayed in Table 1. Percentages missing can be found in Table A1 and Table A3 in Appendix A and Appendix D for each variable. 

### 3.2. Angiographic Features

The most frequently treated vessel was the left anterior descending artery (*n* = 970, 45.9%), followed by the right coronary artery (*n* = 794, 37.6%) and the circumflex artery (*n* = 479, 22.7%). Thrombolysis in myocardial infarction (TIMI)-flow < 3 was present in 87.2% (*n* = 1695) of patients before PCI and in 18.8% (*n* = 375) of patients after PCI. Of all patients with multivessel disease, multivessel PCI was performed in 28% (*n* = 359). A decreasing trend over the years was observed in multivessel PCIs performed in patients with multivessel disease (See Figure 1). Vascular access was achieved through the radial artery in 49.3% and the femoral artery in 50.2% of patients. A temporal trend toward less femoral access was seen over the years (60.7%, 55.6%, 52.6%, 47.8% and 48.5% from 2017 to 2021; *p*-value for trend = 0.019). Overall, unadjusted mortality was significantly higher in the femoral access group (63.3% vs. 36.1%, *p* < 0.001). 

### 3.3. Mechanical and Pharmacological Support

The majority of patients (79.1%, *n* = 1842) received at least one inotropic/vasopressor drug during admission. A total of 710 patients (32.4%) were treated with two vasoactive agents, and ≥3 agents were administered to 494 patients (22.5%). Norepinephrine was the drug most frequently used (70.9%, *n* = 1613) either in combination or not with other drugs, followed by dobutamine (30.9%, *n* = 699) and enoximone/milrinone (20.2%, *n* = 458). Mechanical circulatory support was initiated in 544 patients (23.6%). As demonstrated in Figure 2, this amount was mainly driven by intra-aortic balloon pumps (IABPs).

### 3.4. Survival

The overall 30-day mortality was 38.7% (*n* = 901), and this percentage was stable over the observation period of four years (details are shown in Figure 3). Survival curves for subgroups are shown in Figure 4. The survival rate was higher in patients presenting with STEMI in comparison to NSTEMI (61.8% vs. 53.4%, *p* = 0.005). On average, those presenting with STEMI were younger (67 vs. 69 years, *p* < 0.001) and had lower rates of diabetes (19.1% vs. 31.2%, *p* < 0.001) and prior coronary events (24.2% vs. 51.5%, *p* < 0.001) than those presenting with NSTEMI. In addition to that, the left ventricular ejection fraction at baseline was lower in NSTEMI patients (35% vs 40%, *p* = 0.009), who also presented with multivessel disease more often (76.5% vs. 57.9%, *p* < 0.001). A higher mortality rate was also seen in patients presenting after an OHCA compared to patients who did not experience an OHCA (48.1% vs. 35.4%, *p* < 0.001). The increase in mortality was even higher for cardiac arrests occurring in-hospital (18.5% vs. 9.3%, *p* < 0.001). Mortality at 30 days was higher when revascularization was unsuccessful (TIMI-flow 0 or 1 post-PCI). In patients with multivessel disease, undergoing multivessel PCI was associated with increased mortality. The overall mortality rate at one year was 44.0% (732/1665) with rates ranging from 42.2% to 45.6% for the individual years of index procedures.

## 4. Discussion

We described a real-time reflection of patients with CS who underwent percutaneous revascularization in the Netherlands with national registry data. A total of 2328 shock patients were identified with a mean age of 66.4 years and of whom 72.9% were male. An overall 30-day mortality rate of 38.7% was found. Mortality was higher in patients presenting with NSTEMI compared to patients with STEMI. Higher mortality rates were also seen in patients presenting after an OHCA and in patients who underwent multivessel PCI. Mortality was similar for male and female patients.

A substantial proportion of the observed results paralleled those reported in previous studies, such as the mean age of almost 70 years and the fact that only a small proportion of patients were female. Mean age and gender distribution were as expected based on the existing literature [10,11]. Also, the more generally available baseline values for blood pressure and heart rate were very similar to those found in other CS populations, as well as admission levels of lactate and blood glucose [12,13]. Blood levels of glucose, lactate and hemoglobin have been adopted into several risk-scoring systems for mortality in cardiogenic shock [14,15]. We also found that higher admission levels of glucose and lactate and lower admission levels of hemoglobin were associated with higher mortality. As infarct size is directly correlated to LV function and mortality, it was not surprising to find higher levels of high-sensitive troponin-T and creatine kinase-MB in non-survivors. 

Some remarkable findings were also observed. The reported mortality rate was relatively low compared to general AMICS cohorts that reported mortality rates around 50% [4]. This could partly be attributable to the fact that in this NHR CS cohort, per the definition, all patients underwent PCI, whereas in other cohorts, revascularization rates of around 90% were described [2,4,16]. In addition to revascularization being the only proven effective therapy for AMICS, this could also have led to a more favorable selection of patients who reached the hospital and were in sufficient condition to undergo revascularization [17]. 

Another interesting observation was that mortality was higher in patients presenting with NSTEMI than in patients presenting with STEMI. Previous research on this topic is inconclusive, and survival benefit has been described for both NSTEMI and STEMI etiology of shock [2,4,18]. In this Dutch cohort, demographic features differed between these groups. In general, NSTEMI patients had more severe clinical risk factors, as they were older and had more comorbidities and worse cardiac function at baseline, which could explain the higher mortality rate [19,20]. In our study, we also found that the mortality rate in patients presenting after an OHCA was higher than for non-OHCA patients, which is in line with findings by Ostenfeld et al. but in contrast with other results from Denmark [4,13]. This could again be due to lower revascularization rates in the two latter Danish cohorts than in the current Dutch cohort. As described in the results, in-hospital cardiac arrests (IHCAs) affected mortality more than OHCAs. This phenomenon is not uncommon, and we hypothesized that a higher rate of comorbidities in IHCA patients causes this difference, as this has been described previously [21]. 

Even though the evidence with regard to therapeutic strategies is limited, a few statements have been adopted into the guidelines for the treatment of AMICS. In 2017, multivessel PCI for the index procedure was shown to be associated with a worse outcome than single-vessel PCI in patients with multivessel disease [12]. Although the recommendations from the CULPRIT-SHOCK trial were not clearly seen in the first years after publication, it is evident that in the subsequent years, multivessel PCI during the index procedure was performed less and less in patients with multivessel disease. This could be interpreted as a real-world implementation of new evidence in routine clinical practice. The authors hypothesized that despite the results of the CULPRIT-SHOCK trial, physicians may still feel the need to perform immediate multivessel PCI in case of a lack of hemodynamic improvement after initial treatment of the culprit lesion. 

In the current cohort, the most frequently used vasoactive agent was norepinephrine, which was administered to 71% of patients. This strategy was consistent with both the American and the European recommendation on medical therapy in CS, as norepinephrine is suggested as the first-choice vasopressor [19,20].

Finally, the role of mechanical circulatory support (MCS) in the treatment of AMICS patients remains unclear. Even though a survival benefit for patients treated with MCS has yet to be established, almost one quarter of patients in this cohort were supported with at least one MCS device. After the results of the IABP-SHOCK II trial were published in 2012, the routine use of IABP was no longer recommended by the guidelines [22]. Despite these results, 14.6% (*n* = 337) of patients were treated with an IAPB either in combination or not with another device. Randomized evidence from large trials concerning Impella or veno-arterial extracorporeal membrane oxygenation (VA-ECMO) is not readily available, as trials are still recruiting. Treating physicians may at times feel the need to deploy MCS despite the current lack of evidence for their usage. Even though the incidence of MCS use in the Netherlands seems high, rates of MCS use in other contemporary cohorts are similar, ranging from 19% to 35% [23,24]. The distribution between Impella and VA-ECMO, with Impella being used more often, is comparable with other reports. 

To the best of our knowledge, this is the largest cohort of patients with CS who underwent PCI with data available on clinical, biochemical and angiographic parameters. It provides a real-world insight covering 49% of all CS patients nationwide in the selected timeframe. Data collection was performed with great care, and high standards of quality control as set by the NHR, were applied. In addition to that, patient survival status was retrieved from the governmental Personal Records Database, guaranteeing reliable documentation. Finally, the amount of variables with high percentages of missing data were limited, especially for those variables that are routinely collected in all patients undergoing PCI. 

However, this registry had some limitations as well. Firstly, some selection bias may have been introduced by the partly retrospective aspect of the study. Patients who were initially classified as being in shock but had no source documents confirming the diagnosis of shock other than being labeled as such in the electronic health record, were excluded from the analysis. Nevertheless, this would only strengthen the data on true CS patients. Unfortunately, we did not incorporate the Society for Cardiovascular Angiography and Interventions (SCAI) class definition in our comprehensive CS registry. Regrettably, we did not capture data on bleeding either, which may be of interest, especially in patients treated with mechanical circulatory support.

Furthermore, in some of the additionally collected shock variables, the percentage of missing data exceeded 40%. This was only the case in 5 of these 49 variables, and this was dealt with by providing details on percentages and denominators. 

Lastly, despite applying strict criteria and only including AMICS patients who underwent PCI, some heterogeneity in the population was inevitable. Only AMI-related CS in patients who underwent PCI was included, but associations between risk factors and outcome could vary for different sub-etiologies; e.g., high lactate on admission might be more indicative of a bad prognosis in non-resuscitated patients than in patients presenting after an OHCA. Nevertheless, we believe that the present variety is in fact a strength because it reflects a real-world population. 

## 5. Conclusions

This contemporary Dutch cohort describes characteristics and outcomes of 2328 patients with AMICS undergoing PCI. The all-cause mortality at 30 days was 38.7%. Considerable differences were seen in patient, hemodynamic and biochemical characteristics between survivors and non-survivors. Interestingly, multivessel PCI and IABPs were frequently applied despite currently available evidence. 

## Figures and Tables

**Figure 1 jcm-12-05221-f001:**
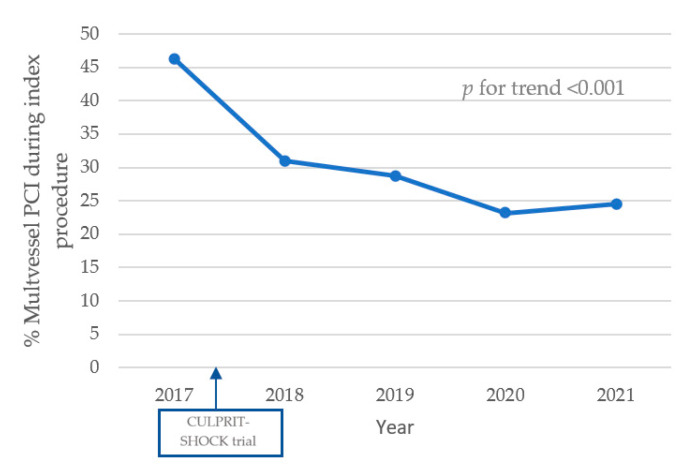
Percentage of multivessel PCIs during index procedure within patients with multivessel disease.

**Figure 2 jcm-12-05221-f002:**
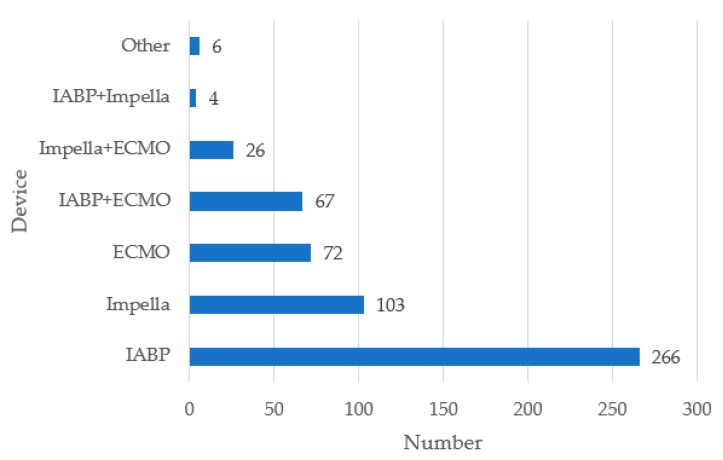
Use of mechanical circulatory support.

**Figure 3 jcm-12-05221-f003:**
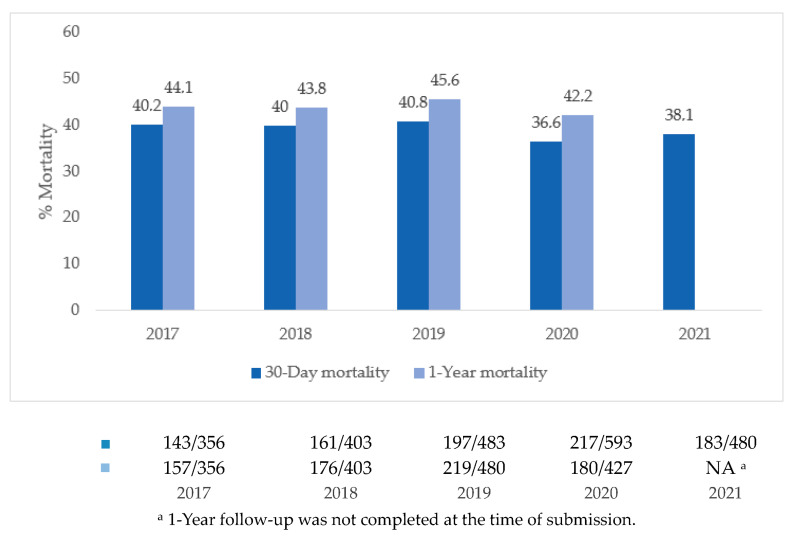
Yearly trend in 30-day and 1-year mortality.

**Figure 4 jcm-12-05221-f004:**
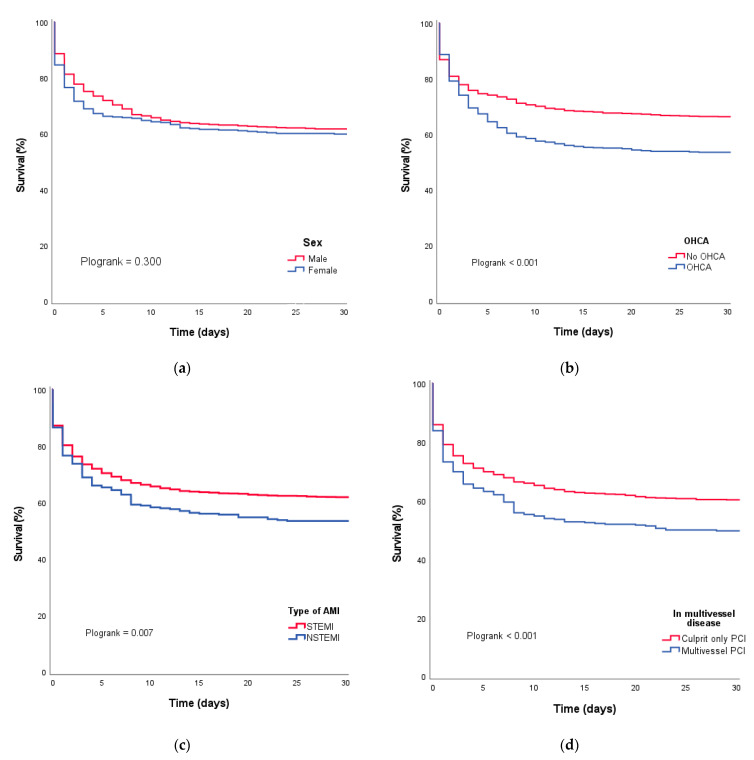
Survival curves for (**a**) males and females; (**b**) OHCA yes or no; (**c**) STEMI and NSTEMI; (**d**) multivessel and single-vessel PCI.

**Table 1 jcm-12-05221-t001:** Patient characteristics for all patients, survivors at 30 days and non-survivors at 30 days.

		All Patients (*n* = 2328)	Alive at 30 Days (*n* = 1414)	Dead at 30 Days (*n* = 901)	*p*-Value
Patient characteristics					
Male		1696 (72.9)	1036 (73.3)	649 (72.0)	0.515
Age—years		66.4 (±12.3)	64.8 (±12.1)	69.0 (±12.1)	<0.001
BMI—kg/cm^2^		26.1 (23.9–29.1)	25.9 (23.7–28.8)	26.2 (24.2–29.4)	0.024
Indication of PCI					0.005
	*STEMI*	1941/2254 (86.1)	1193/1359 (87.8)	737/882 (83.6)	
	*NSTEMI*	313/2254 (13.9)	166/1359 (12.2)	145/882 (16.4)	
Out-of-hospital cardiac arrest		934/2317 (40.3)	497/1405 (35.4)	432/899 (48.1)	<0.001
In-hospital cardiac arrest		295 /2308 (12.8)	130/1401 (9.3)	165/894 (18.5)	<0.001
Onset of AMI symptoms—hours					<0.001
	*<3*	1166/1991 (58.6)	745/1233 (60.4)	416/746 (55.8)	
	*3–12*	375/1991 (18.8)	245/1233 (19.9)	128/746 (17.2)	
	*12–24*	113/1991 (5.7)	67/1233 (5.4)	44/746 (5.9)	
	*>24*	337/1991 (16.9)	176/1233 (14.3)	158/746 (21.2)	
Intubation pre-PCI		1030/2307 (44.6)	500/1404 (35.6)	524/893 (58.7)	<0.001
Monitoring via PA catheter		118/2119 (5.6)	68/1287 (5.3)	49/832 (5.9)	0.613
Medical history					
Diabetes		463/2219 (20.9)	227/1365 (16.6)	232/841 (27.6)	<0.001
Prior coronary event		631/2153 (29.3)	361/1310 (27.6)	265/831 (31.9)	0.032
	*Prior MI*	482/2253 (21.4)	276/1374 (20.1)	202/867 (23.3)	0.071
	*Prior PCI*	396/2134 (18.6)	239/1299 (18.4)	153/822 (18.6)	0.901
	*Prior CABG*	139/2286 (6.1)	74/1390 (5.3)	65/833 (7.4)	0.048
Hemodynamics on admission					
Systolic blood pressure—mmHg		100 (80–125)	103 (83–127)	95 (80–118)	<0.001
Diastolic blood pressure—mmHg		61 (50–77)	64 (50–80)	60 (48–75)	<0.001
Mean blood pressure—mmHg		75 (60–93)	77 (63–95)	72 (58–89)	<0.001
Heart rate—bpm		82 (63–101)	80 (60–100)	89 (70–108)	<0.001
Shock index		0.76 (0.58–1.0)	0.72 (0.56–0.95)	0.86 (0.64–1.14)	<0.001
Number of vasoactive agents pre-PCI					<0.001
	*None*	1147/2215 (51.8)	833/1356 (61.4)	309/846 (36.5)	
	*1*	590/2215 (26.6)	320/1356 (23.6)	267/846 (31.6)	
	*2*	376/2215 (17.0)	171/1356 (12.6)	201/846 (23.8)	
	*≥3*	102/2215 (4.6)	32/1356 (2.3)	69/846 (8.1)	
Laboratory values on admission					
Lactate—mmol/L		5.5 (2.6–9.4)	4.2 (2.1–7.2)	7.8 (3.9–11.4)	<0.001
Creatinine—µmol/L		100 (82–123)	94 (78–113)	110 (91–140)	<0.001
eGFR—mL/min		61 (48–75)	65 (53–80)	54 (40–67)	<0.001
Hemoglobin—mmol/L		8.3 (±1.4)	8.4 (±1.3)	8.1 (±1.5)	<0.001
Glucose—mmol/L		12.2 (8.8–17.1)	10.8 (8.3–14.9)	14.8 (10.4–19.9)	<0.001
Peak hs-troponin-T—ng/L ^a^		3534 (828–10000)	3292 (831–10000)	3954 (772–10000)	0.095
Peak CK-MB—U/L ^a^		222 (70–510)	203 (67–446)	269 (77–600)	0.013
Angiographic features					
Multivessel disease		1402/2307 (60.8)	791 / 1399 (56.5)	603 / 895 (67.4)	<0.001
Number of treated vessels					<0.001
	*1*	1749/2114 (82.7)	1115/1295 (86.1)	623/806 (77.3)	
	*≥* *2*	365/2114 (17.3)	1801295 (13.9)	183/806 (22.7)	
Treated vessel					
	*Left main*	292/2114 (13.8)	142/1295 (11.0)	149/806 (18.5)	<0.001
	*Left anterior descending*	970/2114 (45.9)	576/1295 (44.5)	388/806 (48.1)	0.102
	*Circumflex artery*	479/2114 (22.7)	250/1295 (19.3)	226/806 (28.0)	<0.001
	*Right coronary artery*	794/2114 (37.6)	534/1295 (41.2)	254/806 (31.5)	<0.001
	*Venous or arterial graft*	30/2114 (1.4)	14/1295 (1.1)	16/806 (2.0)	0.103
TIMI flow before PCI					0.721
	*0/1*	1487/1943 (76.5)	905/1189 (76.1)	575/744 (77.3)	
	*2*	208/1943 (10.7)	132/1189 (11.1)	74/744 (9.9)	
	*3*	248/1943 (12.8)	152/1189 (12.8)	95/744 (12.8)	
TIMI flow after PCI					<0.001
	*0/1*	182/1999 (9.1)	54/1255 (4.3)	128/735 (17.4)	
	*2*	193/1999 (9.7)	111/1255 (8.8)	81/735 (11.0)	
	*3*	1624/1999 (81.3)	1090/1255 (86.9)	526/735 (71.6)	
Arterial access					<0.001
	*Radial*	1013/2053 (49.3)	718/1242 (57.8)	288/798 (36.1)	
	*Femoral*	1032/2053 (50.3)	521/1242 (41.9)	505/798 (63.3)	
	*Other*	8/2040 (0.3)	3/1242 (0.3)	5/798 (0.7)	
Outcome					
Length of hospital stay—days		5 (1–12)	10 (2–24)	2 (0–6)	<0.001

Values are *n* (%) or median (25th to 75th percentile). BMI = body mass index; PCI = percutaneous coronary intervention; (N)STEMI = (non-)ST-elevation myocardial infarction; (A)MI = (acute) myocardial infarction; PA catheter = pulmonary artery catheter; CABG = coronary artery bypass grafting; Shock Index was calculated as heart rate/systolic blood pressure; eGFR = estimated glomerular filtration rate; CK-MB = creatine phosphokinase-MB; Vasoactive agents pre-PCI = number of drugs that were administered before PCI (from noradrenaline, adrenaline, dopamine, dobutamine and enoximone/milrinone); TIMI = thrombolysis in myocardial infarction; Length of hospital stay is in days. ^a^ Peak values within 3 days after PCI.

## Data Availability

The data presented in this study were obtained from the Netherlands Heart Registration and are not openly available. Data may be provided upon request.

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
