# Peer review of "Characteristics, Treatment Strategies and Outcome in Cardiogenic Shock Complicating Acute Myocardial Infarction: A Contemporary Dutch Cohort"

_jcm, 2023, doi:10.3390/jcm12165221_

Round 1

Reviewer 1 Report

The data is relevant to a Dutch population and similar to other studies with some unique results. I would encourage including patient status consistent with the SCAI Shock scale. This scale have become frequently used to scale patient outcomes. 

In addition, studies have shown that out of hospital cardiac arrest is a major predictor for poor outcomes which seems less in your population. It might be worth assessing differences to other reports for this variation. 

There is no information on Left ventricular function which might be a part of the worse prognosis in NSTEM CS as other studies have suggested worse baseline LV function and more diffuse CAD.

Adding information on bleeding risk would be of interest as well.

In summary, this is a useful database but the information could be enhanced by greater comparisons to other study populations.

Reviewer 2 Report

The authors examined the details of 2000 patients with CS in the data of national database. Although there were limitations in the database, this is a mega-study. However, what is the new findings of this study? How can the readers get a new finding from this paper? Please, highlight a novel finding from this large cohort.  

Round 2

Reviewer 2 Report

The authors clearly mentioned about the significance of the study. The number of enrolled cohort and the up-dataness were reliable.
